# Diverse biophysical and molecular mechanisms drive phytoplankton sinking in response to starvation

Yanqi Wu[1], Vieyiti K. Kouadio[1☉], Thomas R. Usherwood[1,2☉], Justin Li[1], Margaret Bisher[1], Reshum Aurora[1], Aaron Z. Lam[1], Alice R. Lam[1], Abigail K. R. Lytton-Jean[1], Scott R. Manalis[1,3,4], Teemu P. Miettinen[1]*

**1** Koch Institute for Integrative Cancer Research, Massachusetts Institute of Technology, Cambridge, Massachusetts, United States of America, **2** Harvard-MIT Department of Health Sciences and Technology, Institute for Medical Engineering and Science, Massachusetts Institute of Technology, Cambridge, Massachusetts, United States of America, **3** Department of Biological Engineering, Massachusetts Institute of Technology, Cambridge, Massachusetts, United States of America, **4** Department of Mechanical Engineering, Massachusetts Institute of Technology, Cambridge, Massachusetts, United States of America

☉ These authors contributed equally to this work
* teemu@mit.edu

## Abstract

Marine phytoplankton face eco-evolutionary pressure to regulate their vertical position in the ocean to access light, which is abundant towards the surface, and nutrients, which are found deeper down the water column. All phytoplankton experience gravitational sinking, which can contribute to their vertical migration. However, the biophysical and molecular mechanisms that impact gravitational sinking have not been systematically characterized across taxa and environmental conditions. Here, we combine simulations with measurements of cell mass, volume, and composition to investigate the effects of nutrient availability on gravitational sinking in nine representative unicellular pico- and nanoplankton species. We find that gravitational sinking becomes faster in most species when starved, but the biophysical changes responsible for this vary across species and starvation conditions. For example, the faster sinking of *Chaetoceros calcitrans* is nearly exclusively driven by cell density whereas that of *Emiliania huxleyi* is due to cell volume. On the molecular level, the altered sinking is predominantly attributed to changes in cellular dry contents, rather than water. For example, starch accumulation increases sinking in three green algae species, and lipid accumulation decreases sinking in *Phaeodactylum tricornutum*. Overall, our work reveals that phytoplankton physiology has evolved multiple mechanisms that impact gravitational sinking in response to starvation, possibly to support the vertical migration of the cell.

**Data availability statement:** All numerical data are found in S1 Data, which contains multiple datasheets. The first sheet details the data format and the connection between the data and the manuscript figures. Image analysis code has been deposited to Zenodo (https://doi.org/10.5281/zenodo.17478443).

**Funding:** This work was supported by the National Science Foundation (https://www.nsf.gov/) Award 2319028 to S.R.M. The work was also supported, in part, by the Koch Institute Support(core) grant P30-CA14051 from the National Cancer Institute (https://www.cancer.gov/). The funders did not influence study design, data collection, and analysis, or any other parts of the research.

**Competing interests:** I have read the journal's policy and the authors of this manuscript have the following competing interests: S.R.M. is a co-founder of and affiliated with the companies Travera and Affinity Biosensors, which develop techniques relevant to the research presented. The other authors declare that they have no competing interests.

**Abbreviations:** CCMP, Culture Collection of Marine Phytoplankton; PBS, phosphate-buffered saline; SEM, scanning electron microscopy; SMR, suspended microchannel resonator; TEM, transmission electron microscopy.

## Introduction

Phytoplankton are key primary producers in the oceans that support marine food webs and drive carbon fixation [1,2]. Their growth and fitness depend on photosynthesis-derived energy and seawater nutrients–resources that are unevenly distributed in the ocean: light is more abundant towards the surface, while nutrients are more concentrated deeper in the water column [3,4]. This generates eco-evolutionary pressure for phytoplankton to migrate vertically in the water column to meet their energy and nutrient requirements [5]. Nutrient-limited phytoplankton may sink deeper in the water column in search of nutrients, simultaneously contributing to the downward flux of organic carbon. Although it is unclear if phytoplankton regulate their sinking to reach more nutrients, thus achieving a fitness advantage, or if changes in cell sinking are simply byproducts of other metabolic changes, the vertical movement of cells in the ocean has ecological consequences. The vertical movement of phytoplankton is predicted to have a significant impact on primary production and nutrient cycles in the oceans [2,6–8], and field observations have confirmed the sinking of single cells and small particles (<100 μm cell aggregates) as contributors to ocean carbon fluxes [9–12]. Yet, the degree to which phytoplankton sinking is impacted by nutrient limitations, as well as the mechanisms responsible for such phenotypic response, have not been systematically characterized across taxa (Fig 1A).

Here, we focus on gravitational sinking as a potential mechanism for vertical migration. While motile plankton can achieve faster migration with phototaxis than with gravitational sinking, gravitational sinking is experienced by all species and may act as an energy-efficient migration mechanism. The gravitational sinking velocity ($v_{sink}$, from here on referred to simply as sinking) of a cell can be derived from Stokes' law [13–15],

$$v_{sink} = \frac{2\left(\rho_{cell} - \rho_{fluid}\right) g r^2}{9\mu}\, \Phi$$

(1)

where $\rho_{cell}$ is the density of the cell, $\rho_{fluid}$ is the density of the surrounding fluid, $g$ is the gravitational acceleration, $r$ is the equivalent spherical radius of the cell, $\mu$ is the dynamic viscosity of the fluid, and $\Phi$ is the correction factor for non-spherical shape [14,16]. Biophysical properties of a cell, namely cell volume, density, and shape, define sinking (Fig 1B), and could determine the sinking of cell aggregates similarly. At the molecular level, we model a cell as the collection of all its intracellular molecules, where the cell volume is the sum volume of individual molecules, and the cell density is the weighted average of the densities of those molecules (Fig 1B). These molecules can be separated into water and dry contents of the cell. For most phytoplankton, especially those without large silica and calcium carbonate shells/walls, most cellular dry contents fall into three groups—proteins, lipids, and carbohydrates, each with distinct densities (proteins—1.35 g/mL, lipids—0.92 g/mL, carbohydrates—1.5 g/mL, S1 Table) [17,18]. Previous works have used similar models of cell composition [19], and here it provides a framework that relates cell's molecular content to its biophysical properties and sinking (Fig 1B).

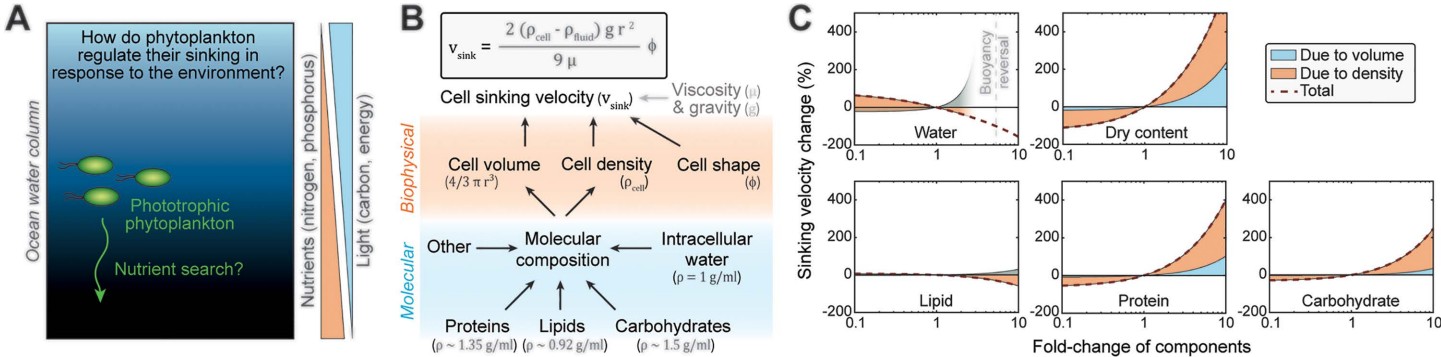

**Fig 1. Gravitational sinking velocity can be simulated using a simple model of cell composition.** **(A)** Schematic of the study question. **(B)** A simple physical model linking cell composition to cell sinking. Gravitational sinking velocity is determined by Stokes' law (top). On biophysical level, cell sinking is dependent of cell volume, density, and shape (middle). On molecular level (bottom), cell volume and density are primarily dependent on cellular water, protein, lipid, and carbohydrate content. **(C)** Simulations of cell sinking velocity in a representative diatom (*Phaeodactylum tricornutum*) as a function of cell composition. Sinking velocity changes due to cell volume and cell density are separated in blue and orange, respectively. Changes in dry content refer to corresponding changes in all other contents except water. The simulation data can be found in the S1 Data file.

Studies of cell sinking using sedimentation columns or timelapse imaging have shown that many species alter their sinking in response to environmental conditions, such as starvation [20–25]. However, the mechanisms responsible for these sinking changes are, in most cases, unknown. Many phytoplankton have also been shown to change their molecular composition in response to starvation [26–30]. Such compositional changes could alter cell sinking, as also suggested by modeling studies [19], but there is limited experimental evidence that quantitatively links changes in cell size and composition to cell sinking. Notable exceptions to this are the species *Pyrocystis noctiluca*, which can undergo long vertical migration due to water content regulation [31], and *Tetraselmis sp.* which can sink faster when starved due to decreased cellular water content and increased carbohydrate content [32].

Here, we study how gravitational sinking responds to different nutrient limitations across a range of motile and non-motile unicellular eukaryotic marine phytoplankton. Using simulations and experiments, we connect cell sinking to the regulation of cell's biophysical properties and molecular composition. Our work reveals multiple mechanisms responsible for starvation-induced sinking in different phytoplankton species, suggesting that phytoplankton may have evolved several solutions to support their vertical migration.

## Results

### Simulation of phytoplankton sinking velocity

To understand how cell sinking could be impacted by molecular composition, we simulated sinking for cells with different compositions (Fig 1B). Our simulations relied on the previously reported macromolecular composition and size of a typical green algae (*Dunaliella tertiolecta*, S1A Fig, S2 Table) or a typical diatom (*Chaetoceros calcitrans*, S1B Fig, and *Phaeodactylum tricornutum*, Fig 1C). We then systematically varied the amount of each major intracellular component (lipid, carbohydrate, protein, and water) while keeping other components constant. In addition, we varied all components except water simultaneously in order to examine the effect of dry contents on cell sinking. For each simulated variation, we determined the fraction of the sinking change caused by cell volume and cell density using a first-order Taylor linear approximation. The results revealed that: (i) faster sinking is more readily achieved by gaining cellular dry contents, specifically carbohydrates and proteins, rather than by decreasing water or lipid content, (ii) change in any single cellular component will alter sinking more due to changes in cell density than changes in cell volume, and (iii) the reversal of cell buoyancy (decreasing cell density below that of seawater) requires dramatic molecular changes such as large accumulation of water

(>4-fold) or lipids (>10-fold) without additional protein or carbohydrate accumulation. These conclusions were not sensitive to the cell composition differences observed between species (S1 Fig). We note that our model groups the rest of cellular contents as "other" and, due to the complex nature of this "other" group, we assume it to be constant in density and volume.

## Phytoplankton increase sinking velocity in response to starvation

We have previously established an approach for determining the sinking of pico- and nanoplankton species by applying Stokes' law to single-cell measurements of cell mass and volume [32–34]. In this approach, we measure single-cell volumes using Coulter counter, single-cell buoyant masses using a suspended microchannel resonator (SMR), and cell shapes using microscopy. To examine starvation-induced changes in cell sinking across the tree of life, we selected nine phototrophic unicellular eukaryotic marine pico- and nanoplankton species and we cultured the cells under photoautotrophic conditions in both high and low nutrient (i.e., starvation) media (Fig 2A, S3 Table). Following a 5-day culture, sinking velocities of a cell population were inferred from population averages of hundreds of single cell volume and buoyant mass measurements (S2A and S2B Fig). We also confirmed that low nutrient condition resulted in starvation (S3A Fig), and that cell shapes did not change between conditions (S3B Fig). Overall, our analysis focused on species which did not display

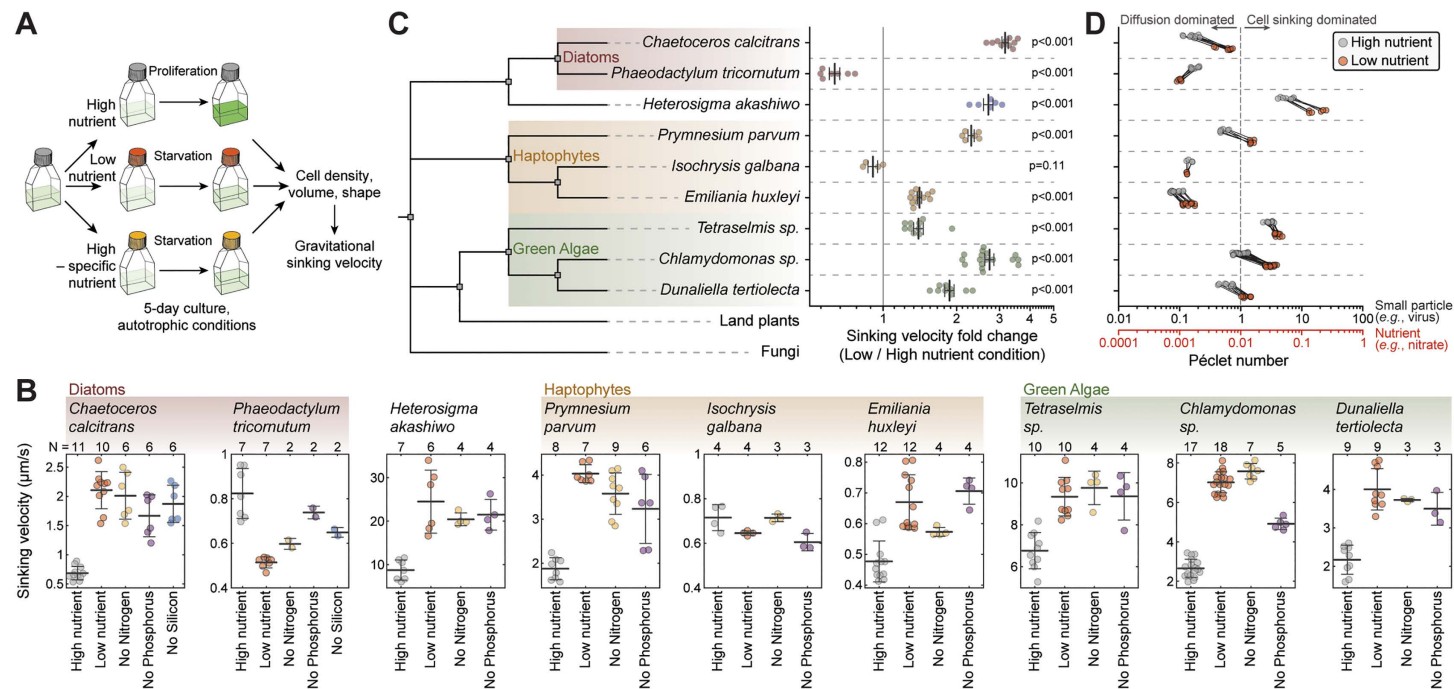

**Fig 2. Cell sinking increases in response to starvation across many genetically distinct, unicellular marine phytoplankton. (A)** Schematic of experimental setup. **(B)** Gravitational sinking velocity in indicated species following 5-day culture under indicated nutrient conditions. N depicts the number of independent cultures (dots), bar and whiskers depict mean±SD. **(C)** Phylogenetic tree of the studied species and the relative sinking velocity change between low (starving) and high (proliferating) nutrient conditions in each species. Land plants and fungi are show for reference. Same data as in (B), dots depict separate cultures, bar and whiskers depict mean±SEM, p-value obtained by t test comparison to the value of 1. **(D)** Péclet numbers for high (gray dots) and low nutrient (orange dots) conditions, as calculated for the diffusion of nitrate (red x-axis) or a virus particle (black x-axis). Species are in the same order as in (C) separated by dotted horizontal lines. Note that paired high and low nutrient condition replicates are connected by a solid line. Péclet number > 1 indicates that the encounter rate of the cell and the particle (nitrate or virus) is dominated by cell sinking rather than particle diffusion. All data can be found in the S1 Data file.

extensive aggregation, and our data reflects single cells that are either proliferating (high nutrient condition) or starving (low nutrient condition).

Most tested phytoplankton species (7 out of 9) displayed increased sinking when starved for nutrients (Fig 2B and 2C). For example, in the diatom *C. calcitrans*, the toxic raphidophyte *Heterosigma akashiwo*, and the green alga *Chlamydomonas sp.*, sinking increased approximately 3-fold under low nutrient conditions compared to high nutrient conditions. In *H. akashiwo*, the largest species studied, sinking reached up to ~25 μm/s when starved. We also observed one diatom species, *P. tricornutum*, that displayed decreased sinking upon starvation. Only the haptophyte *Isochrysis galbana* did not change sinking significantly following starvation (Fig 2C), and this species was excluded from future analyses. Overall, our results indicate that most marine phytoplankton alter their sinking in response to a general nutrient starvation, but the effect magnitude is species-specific even within phylogenetic clades.

### Starvation increases phytoplankton Péclet numbers, promoting small particle encounters

In environments where nutrients are sparse, nutrient uptake is limited by the rate of diffusion. However, cells can promote their nutrient acquisition by movement. To evaluate if the observed sinking changes are sufficient to promote nutrient acquisition, we calculated the Péclet number (Pe) for a representative small nutrient (nitrate). Pe represents the relative contribution of diffusion and cell sinking to mass transport, with $Pe \ll 1$ indicating that diffusion-dominated transport, and $Pe \gg 1$ indicating sinking-dominated transport. Although Pe increased up to 4-fold under low nutrient conditions, the Pe for nitrate remained very small (Fig 2D), indicating that cell sinking does not meaningfully increase nutrient acquisition in the species studied here. We note that this only reflects short timescales when the local nutrient environment remains constant, and over long timescales cells can sink into deeper, nutrient-richer waters. On the other hand, particles, such as viruses or other cells, diffuse significantly more slowly than nutrients, leading to $Pe_{virus} \geq 1$, for many species when starved (Fig 2D). For example, the toxic haptophyte *Prymnesium parvum* shifted from a diffusion-dominated regime of viral encounters ($Pe_{virus} = 0.53 \pm 0.03$, mean ± SEM) to convection-dominated regime of viral interactions ($Pe_{virus} = 1.50 \pm 0.04$, mean ± SEM) when starved. Thus, the sinking we observe may increase cell encounters with particles, such as viruses, other cells, or marine snow.

### Sinking velocity is responsive to multiple nutrients

Next, we examined whether the starvation-induced changes in cell sinking were specific to limitation of a single nutrient. More specifically, we used high-nutrient media that lacks either nitrogen, phosphorus, or silicon (only for diatoms), and we verified that these conditions result in decreased proliferation (S3A Fig). For all species, we observed qualitatively similar results between starvation by overall low nutrient level and by specific nutrients (Fig 2B). However, in some species, the magnitude of sinking change varied between specific nutrient starvations. For example, the sinking change in *C. calcitrans* was greater when starved for nitrogen ($p = 0.016$, paired $t$ test) or silicon ($p = 0.003$, paired $t$ test) than when starved for phosphorus. In contrast, in *Emiliania huxleyi*, sinking increased more when starved for phosphorus than nitrogen ($p = 0.005$, paired $t$ test). Overall, cell sinking is responsive to multiple nutrients, although in some species the effect size varies between different limiting nutrients.

To examine whether the observed changes in sinking were caused by cell death, which can increase cell density [34], we resupplied nutrients to the starved cultures in a subset of species studied. The results revealed that sinking could be rescued with nutrient resupply (S4 Fig). This is consistent with previous work examining cell size and proliferation recovery following starvation in *Tetraselmis sp.* and *E. huxleyi* [32,35].

### Biophysical mechanisms for sinking velocity changes are species-specific

Cell sinking is defined by cell volume, density, and shape, the last of which did not change between conditions (S3B Fig). Here, we consider cell volume as an indicator of total cellular contents, cell density as an indicator of cell composition

(Fig 1B), and we examine how sinking is impacted by cell volume and density. We overlaid the measured cell volume and density changes with the resultant sinking (Fig 3A). This revealed that cell density increased in nearly all species upon starvation, except *E. huxleyi*, which displayed a constant density, and *P. tricornutum*, which decreased cell density. Many phytoplankton species also increased their cell volume upon starvation, as has been previously reported for *E. huxleyi* [35].

We then sought to decouple cell density and volume from Eq (1) using the first-order Taylor approximation (see Materials and methods) in order to determine their relative contributions to cell sinking under each starvation condition. This revealed that green algae, i.e., *Tetraselmis sp.*, *Chlamydomonas sp.*, and *D. tertiolecta*, as well as *H. akashiwo*, relied predominantly on cell density regulation to adjust their sinking (Fig 3B). In contrast, *P. tricornutum* and *P. parvum* relied approximately evenly on cell volume and density regulation to adjust sinking, whereas *E. huxleyi* relied exclusively on cell volume regulation to adjust sinking. Thus, both cell volume and density regulation can function as the biophysical basis for adjusting sinking, depending on the species.

Starvation for different nutrients resulted in different cell volume and density responses. Most notably, in *C. calcitrans*, nitrogen starvation increased cell density with little effect on cell volume, whereas silicon starvation increased cell volume rather than cell density (Fig 3A and 3B). A broader comparison across all species revealed that nitrogen starvation was more prone to alter cell sinking due to cell density, when compared to phosphorus starvation (Fig 3C).

### Increased cell sinking is driven by dry content accumulation

In theory, the observed changes in sinking, as well as biophysical properties, must be attributed to changes in specific cellular molecules. Our simulations of cell sinking suggested that an overall accumulation of cell dry contents can

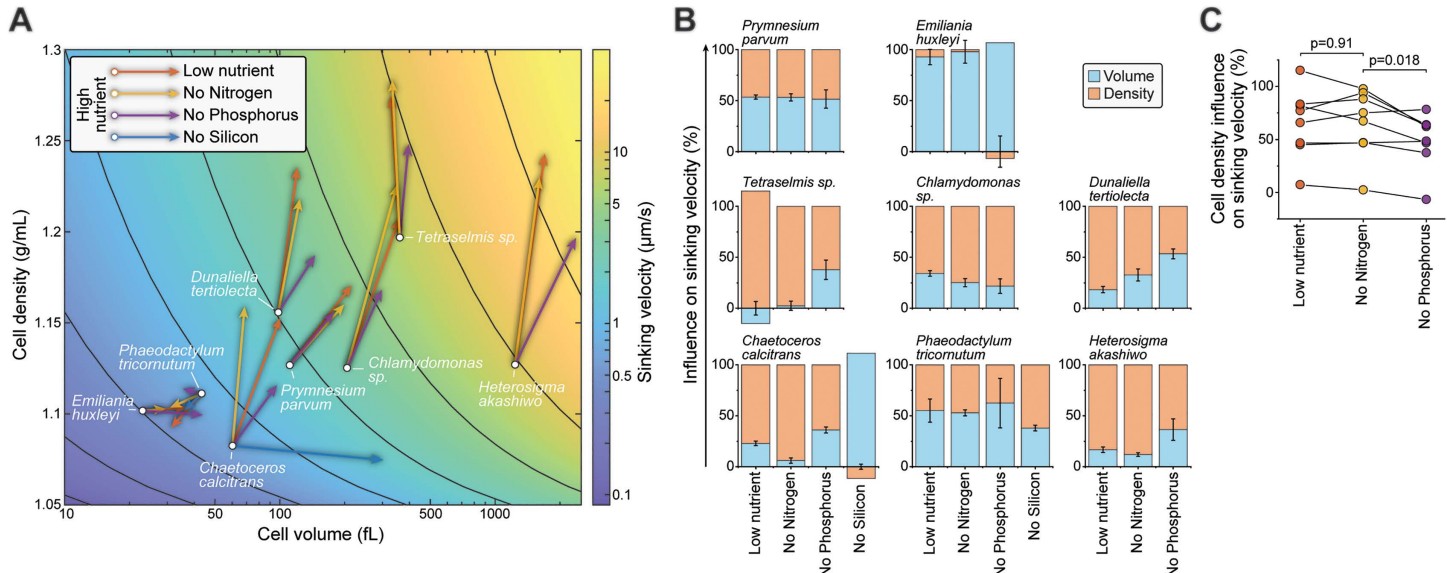

**Fig 3. Biophysical changes responsible for starvation-induced cell sinking. (A)** Changes in cell density and cell volume when comparing high nutrient condition (base of each arrow) to indicated starvation conditions (tip of each arrow). Gravitational sinking velocities are indicated in color-coded background. **(B)** The relative influence of cell volume (blue) and cell density (orange) changes on cell sinking velocity changes under each starvation condition. Negative values imply that volume and density changes have opposite effects on the sinking velocity. Data depicts mean±SEM. **(C)** The relative influence of cell density changes on cell sinking velocity changes under indicated starvation conditions. Dots depict different species, data from same species are connected by a line. *p*-value obtained by Student *t* test (*N* = 8 species). Low nutrient condition and nitrogen starvation appear similar, most likely because nitrogen is the limiting nutrient for most species under the low nutrient conditions. All data can be found in the S1 Data file.

increase sinking more effectively than the loss of water or lipids (Fig 1C). Motivated by this, we sought to examine if the starvation-driven increases in sinking were driven by dry content accumulation or by loss of intracellular water. We used a previously established approach [32], where cells' buoyant masses are determined in normal and heavy water to determine the dry volume of the cells (*i.e.*, dry content), which can then be compared to the total volume of the cells to determine the water volume (*i.e.*, water content). Most species increased their water and dry content following starvation, except for low nutrient starved *C. calcitrans* and *Tetraselmis sp.*, and nitrogen starved *Chlamydomonas sp.* (S5A Fig). These results suggest a separate regulation of phytoplankton's water and dry contents, as observed in other model systems [36]. We note that *H. akashiwo* was excluded from this and future experiments due to technical reasons.

We then compared how cellular water and dry content changes influence cell sinking using a Taylor expansion-based approach (see Materials and methods). This revealed that nearly all starvation-driven sinking increases were caused by increases in cellular dry contents (Fig 4). In contrast, cellular water content, which increased in most species following starvation, had a negative influence on sinking (Fig 4). Only in low nutrient starved *Tetraselmis sp.* did the loss of cellular water contribute positively and significantly to the sinking increase ($p = 0.024$, one-sample $t$ test). Therefore, while starvation induced species- and condition-specific changes to cellular water content, cellular water content was rarely an important contributor to sinking. Instead, the increases in sinking were nearly exclusively driven by changes in cellular dry contents. For a breakdown of cellular dry content into its volume and density, and their separate impact on sinking, see S5 Fig.

### Increased lipid and decreased protein content explain the decreased cell sinking in starving *Phaeodactylum tricornutum*

To further understand which macromolecules impact cell sinking following starvation, we examined the cell lipid and protein compositions. We first focused on explaining sinking changes in *P. tricornutum*, the only species in our study that decreased sinking following starvation. We fluorescently labeled neutral lipids, which are the principal form of storage lipids [37], and cellular proteins in fixed cells and measured the cells using flow cytometry. Lipid accumulation was significantly larger in *P. tricornutum* than in other species tested (S6A Fig). Starved *P. tricornutum* cells contained ~10-fold more

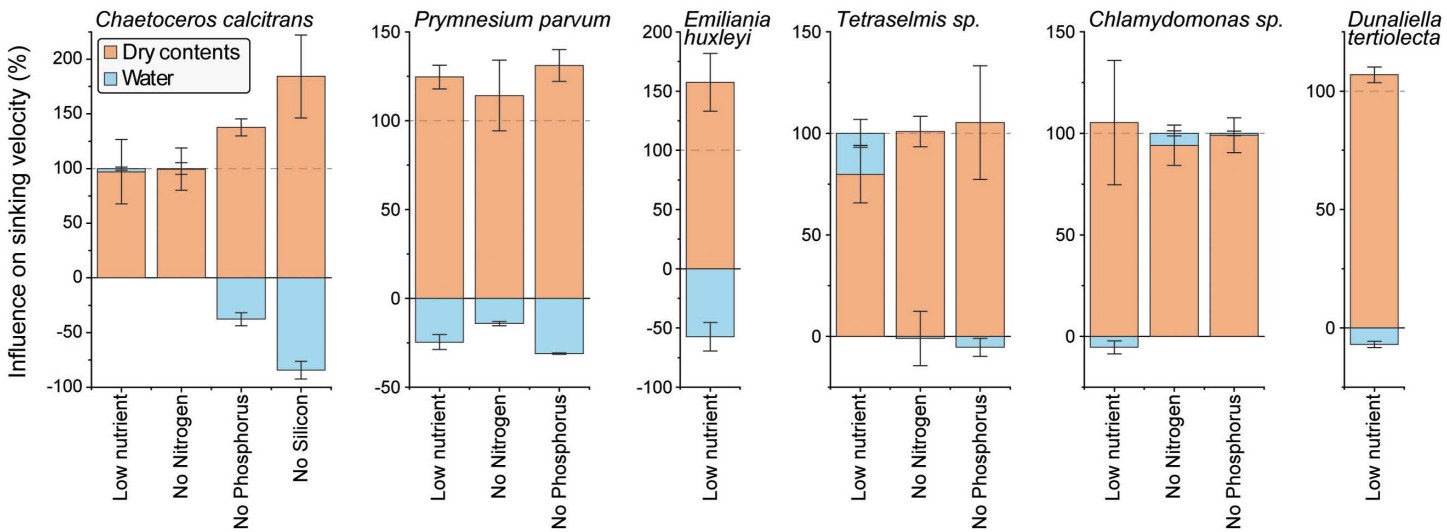

**Fig 4. Starvation-induced cell sinking is driven by dry content accumulation.** The relative influence of cellular water (blue) and dry contents (orange) changes on cell sinking velocity changes under each starvation condition. Negative values imply that dry and water content changes have opposite effects on the sinking velocity. Data depicts mean ± SEM. All data can be found in the S1 Data file.

lipids than cells under high nutrient condition (Fig 5A), consistent with previous reports establishing this species as a high lipid producer [37,38]. Fluorescence and transmission electron microscopy (TEM) revealed that the neutral lipids formed 1–2 large lipid droplets under starvation, whereas the lipids were distributed into smaller droplets under the high nutrient condition (Fig 5B and 5C). We did not observe obvious changes in the frustule thickness (Fig 5C). When analyzing cellular protein content, we found that *P. tricornutum* displayed a larger decrease than other species did when starved (S3B and S6B Figs), with protein content decreasing ~4-fold (Fig 5D). According to our simulations (Fig 1C), the lipid accumulation and protein loss following starvation in *P. tricornutum* are both sufficient to explain the decreased sinking (Fig 5E). However, as the combined effect exceeds the experimentally derived sinking, additional compositional changes must also exist (Fig 5E).

We then examined if we could rescue the sinking decrease in *P. tricornutum* by preventing lipid accumulation. We treated cells with two fatty acid synthase inhibitors, cerulenin and C75, for the duration of the low nutrient starvation. We then imaged the cells for neutral lipids (Fig 5B), and quantified neutral lipid content, cell density, volume, and sinking (Fig 5F). Cerulenin partly reversed the lipid accumulation and lipid droplet morphology observed under low nutrient starvation,

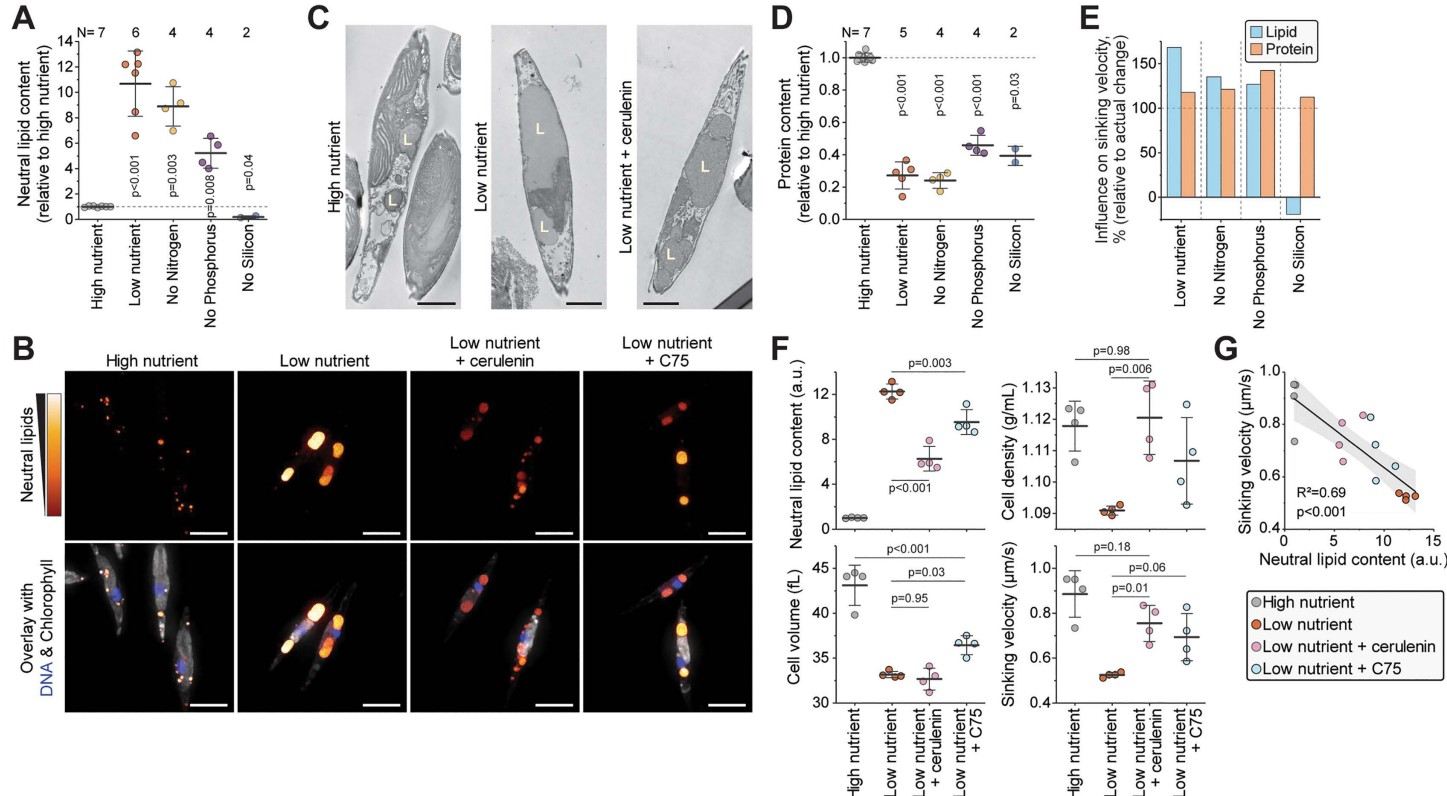

**Fig 5. Extensive lipid accumulation and proteins loss explains the decreased sinking velocity in starving *Phaeodactylum tricornutum*. (A)** Relative changes in cellular neutral lipid content following 5-day culture under indicated starvation conditions. N depicts the number of independent cultures (dots), bar and whiskers depict mean±SD. **(B)** Fluorescence imaging of neutral lipids (red to yellow), DNA (blue), and chlorophyl (gray) following 5-day culture under indicated conditions. Scalebars depict 5 µm. $n > 40$ cells per condition. **(C)** TEM images following 5-day culture under indicated starvation conditions. Scalebars depict 2 µm. Letter L indicates lipid droplets. $n > 25$ cells per condition. **(D)** Same as (A), but data is for cellular protein content. **(E)** Simulated influence of lipid and protein content changes on cell sinking velocity. Data is normalized to the actual sinking velocity changes observed. **(F)** Neutral lipid content, cell density, volume, and sinking velocity of cells under indicated conditions. Dots depict separate cultures ($N = 4$), bar and whiskers depict mean±SD, *p*-value obtained by Student *t* test. **(G)** Correlation between cell neutral lipid content and sinking velocity in the data in (F). Data in all panels is from the species *P. tricornutum*. All data can be found in the S1 Data file.

as was cell sinking velocity (Fig 5B, 5C, and 5F). Cerulenin fully rescued the cell density decrease observed under low nutrient state, but it did not rescue the cell volume decrease. C75 treatment yielded more modest rescues of lipid content and cell sinking than the cerulenin treatment. Overall, the neutral lipid content and sinking of the cells treated with and without fatty acid synthase inhibitors were correlated ($p < 0.001$, ANOVA, $R^2 = 0.69$) (Fig 5G), indicating that a majority of the low nutrient starvation-driven sinking can be attributed to lipid accumulation.

Other species, including *P. parvum* and *Chlamydomonas sp.*, also increased cellular lipid content when starved (S6A Fig), despite increased cell density (Figs 3A and S3A). The lipid accumulation in *Chlamydomonas sp.* was localized exclusively to the cell periphery, where lipid droplets protruded against the plasma membrane (S7A and S7B Fig). This cellular organization is different from the widely studied freshwater counterpart *Chlamydomonas reinhardtii* [39]. These lipid droplets increased 2-fold in diameter, but not in number, upon starvation (S7C and S7D Fig). In addition, *Chlamydomonas sp.* decreased its protein content by ~2-fold when starved. These lipid and protein content changes are expected to decrease *Chlamydomonas sp.* sinking. As we did not observe this experimentally, we expect additional compositional changes to counteract the lipid increase and protein loss in *Chlamydomonas sp.*

### Increased starch reservoirs can explain the increased cell sinking in starving green algae

Green algae are capable of accumulating both starch and lipid reservoirs under starvation, as carbon fluxes are directed from biosynthesis to storage molecules [28,29]. However, the accumulation of starch and lipids would have opposing effects on cell sinking (Fig 1C). This motivated us to examine starch accumulation in green algae. We carried out TEM imaging of the *Tetraselmis sp.*, *Chlamydomonas sp.*, and *D. tertiolecta* (Figs 6A and S8A). We observed extensive starch granule accumulation in all three species when starved, with up to ~40% of the cell area being occupied by starch in *D. tertiolecta* (Fig 6B). This starvation-induced increase in starch content was due to both the number and size of the starch granules increasing (S8B and S8C Fig). In addition, we observed that phosphorus-starved *Chlamydomonas sp.* cells

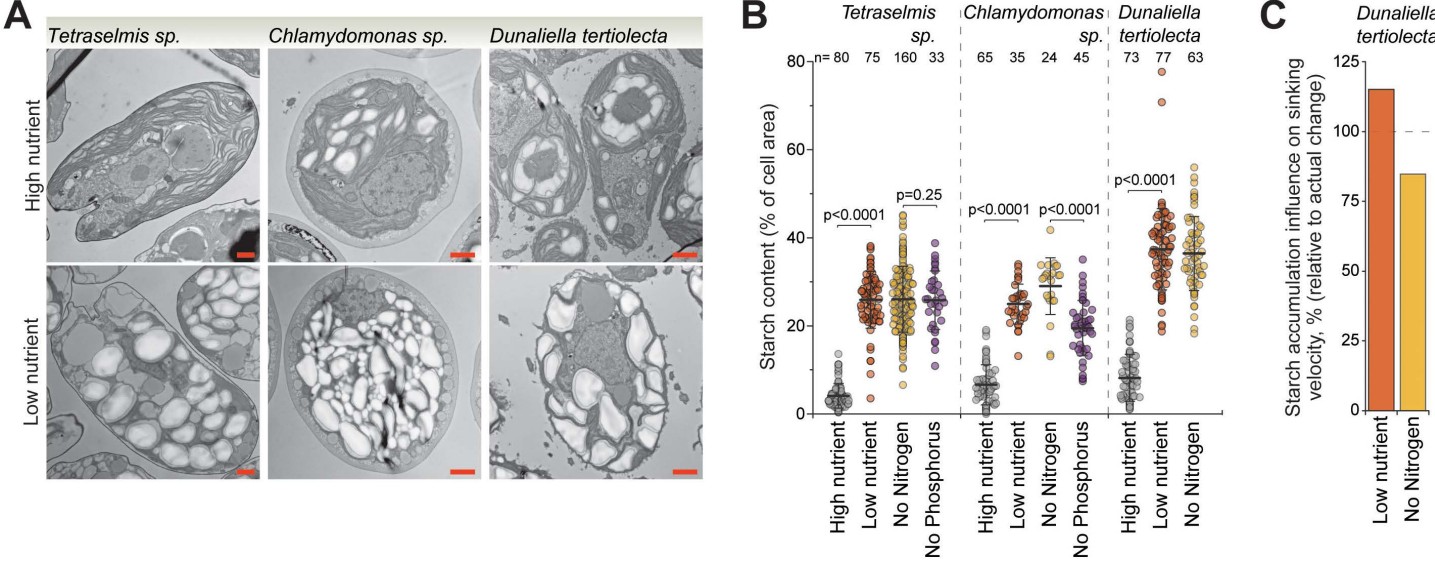

**Fig 6. Extensive starch accumulation can explain sinking velocity increases in starving green algae. (A)** TEM imaging of green algae species following 5-day culture under indicated starvation conditions. Scalebars depict 1 μm. **(B)** Quantifications of cellular starch granule content from TEM images. Dots depict separate cells, bar and whiskers depict mean±SD, *n* depicts the number of cells measured, and *p*-values were obtained using ANOVA followed by Tukey's posthoc test. **(C)** Simulated influence of starch accumulation on cell sinking velocity in *Dunaliella tertiolecta*. Data is normalized to the actual sinking velocity changes observed. All data can be found in the S1 Data file.

accumulated less starch than nitrogen-starved cells, although this difference was not observed in *Tetraselmis sp.* (Fig 6B). Instead, in *Chlamydomonas sp.*, phosphorus starvation resulted in the appearance of acidocalcisome-like organelles [40], which may also contribute to sinking (S8A and S8D Fig).

We used *D. tertiolecta* as a model system for simulations of the influence of starch accumulation on sinking. We chose this species because it displayed little lipid or fractional water content changes when starved (S5 and S6 Figs), suggesting that sinking may rely more exclusively on carbohydrate accumulation. According to our simulations, the accumulation of starch alone explains approximately all the sinking increase observed in low nutrient starved and nitrogen starved *D. tertiolecta* (115% and 85%, respectively) (Fig 6C). More broadly, across all three green algae species, the starch accumulation had a larger simulated impact on sinking than changes in cell lipid, protein, or water content (S8E Fig). Thus, our results indicate starch accumulation as an important molecular mechanism responsible for green algae sinking under nutrient starvation.

## Discussion

Our study revealed that 8 out of 9 tested phytoplankton species alter their sinking when starved for nutrients. In most tested species, this can be explained by the accumulation (or loss) of dry contents, and we have exemplified how the accumulation of lipids and carbohydrates, and the loss of proteins, can function as mechanisms to alter cell sinking. However, it is important to recognize that additional mechanisms that influence sinking can also exist. Some phytoplankton can accumulate large amounts of pigment molecules [30], polyphosphate storages [40], or inorganic components, such as silica and calcium carbonate, all of which may increase cell sinking. Curiously, our experiments did not reveal any conditions where phytoplankton become buoyant in seawater. As shown by our simulations, this would require a large accumulation of water in the absence of other dry content accumulation, or a very large accumulation of lipids. Both mechanisms are likely to require extraordinary changes to intracellular organization, as shown in *Pyrocystis noctiluca* [31], and may therefore be rare. Alternatively, phytoplankton would have to lose >70% of their dry contents, which would likely compromise many cellular functions, or accumulate extremely low-density contents, such as gas vesicles, which are currently only reported to exist in prokaryotes [41,42]. Thus, while our work illustrates several mechanisms used to increase eukaryotic phytoplankton sinking, mechanisms that achieve buoyancy are still largely unknown. More broadly, it is important to note that the in situ vertical migration of phytoplankton is under significant influence by turbulence and ocean currents, and the importance of gravitational sinking of single cells to the movement of phytoplankton remains an open question.

An unexpected discovery in our study is that many phytoplankton grow larger when starved. While the increased cell size is easily explained by decreased proliferation rates [43], it is in a stark contrast to most model systems, where cell size decreases with nutrient starvation [34,44,45]. Why would phytoplankton have evolved to increase their size upon starvation? Increased cell sinking is one possible explanation, but not the only one, especially as sinking was predominantly driven by density rather than volume changes. If phytoplankton sink significantly deeper in the water column, increased cell size can store more energy (lipids and carbohydrates), which may support cell viability and motility in the light-limited deeper waters. A better understanding of the size-dependence of metabolic processes within a species will help elucidate this [46].

We also find that most phytoplankton are denser when starved than when proliferating. This finding resembles observations in several other model systems from bacteria to humans [34,47–49]. This suggests that there may exist a more fundamental "starvation state" where cellular properties are adjusted to cope with starvation, possibly to conserve energy [48]. We did not observe systematic cell size increases under starvation, suggesting that this starvation state is distinct from that observed upon genome dilution, where cells enter starvation-like state due to excessive cell size increases in the absence of DNA replication [50,51]. Importantly, we also identified a few exemptions to this behavior, such as the starvation of *E. huxleyi*, where density did not increase. Future studies comparing these differential starvation responses may elucidate the physiological consequences of the high-density starvation state.

It is possible that the changes in cell sinking that we observe following starvation do not necessarily reflect regulation of cell sinking, but rather byproducts of starvation-induced metabolic effects. The gravitational sinking velocity of most species studied here is under 1 m/day, which makes long-distance (*e.g.*, 50 m) vertical migration very slow and unlikely, especially for small cells. However, in a competitive environment, a fitness advantage might be gained by much more modest changes in depth. This is especially true for cells that inhabit depths close to the nutricline. In addition, gravitational sinking could be augmented by motility and cell aggregation, which were not studied here, but could also change in response to starvation. Our results also suggest that the increased cell sinking velocity following starvation could promote cell-to-cell encounters, possibly promoting cell aggregation and thus sinking.

Finally, our results could support the modeling of ocean ecosystems and nutrient cycles. Our study has connected the macromolecular content of cells to their sinking, and the macromolecular content is also indicative of the C:N:P ratio of cells [30,52]. It seems therefore likely that cells, as well as cell aggregates, with different elemental ratios sink at different rates, which could contribute to marine carbon and nutrient cycles. Linking elemental stoichiometry and macromolecular content to the vertical movement of cells and cell aggregates is an important area of future modeling efforts.

## Materials and methods

### Simulations of cell sinking velocity

The gravitational sinking velocity of a cell is a function of cell size, density, and shape, as described in Eq (1). All phytoplankton species in this study can be modeled as spheroid or ellipsoid (S3 Fig), with a multiplicative shape correction factor less than 10% compared to a sphere [53]. Thus, cell shape was not considered in the simulation, and all cells were assumed to be spherical.

We modeled a cell as the collection of all its intracellular molecules and categorized these molecules into five groups: proteins, lipids, carbohydrates, water, and others. Hence, the cell volume is the sum volume of individual molecules, $V = \sum V_i$, and the cell density is the weighted average of the densities of those molecules (S1 Table), $\rho = \sum w_i \cdot \rho_i$, where the subscript $i$ refers to each molecule group and $w_i$ refers to their volume fraction. Once the cell volume and density are defined by the molecular composition, the sinking velocity can be calculated using Eq (1) for any given cell state.

Using our model, we simulated sinking velocities of *P. tricornutum*, *D. tertiolecta*, *C. calcitrans*, and a hypothetical average species. We first determined their molecular compositions under high nutrient conditions from literature values (S2 Table). For each species, we then varied the volume of one molecule group at a time or, in the case of "dry contents," all non-water molecule groups together, while the other parameters remained constant.

### Influence of biophysical properties and intracellular molecules on sinking velocity

To determine the influence of biophysical properties (i.e., cell volume and density) on cell sinking, we used a first-order Taylor expansion near the baseline condition $(\rho_0, r_0)$ that represents high nutrient condition of the cell.

$$v_{\text{sink}} = f(\rho, r) \approx f(\rho_0, r_0) + \frac{\partial f(\rho_0, r_0)}{\partial \rho} \cdot (\rho - \rho_0) + \frac{\partial f(\rho_0, r_0)}{\partial r} \cdot (r - r_0)$$

Hence, the influence of cell density and volume, $I_{\text{density}}$ and $I_{\text{volume}}$, respectively, can be defined as the fractions of the two derivative terms.

$$I_{\text{density}} = \frac{\frac{\partial f(\rho_0, r_0)}{\partial \rho} \cdot \Delta\rho}{\frac{\partial f(\rho_0, r_0)}{\partial \rho} \cdot \Delta\rho + \frac{\partial f(\rho_0, r_0)}{\partial r} \cdot \Delta r}$$

$$I_{\text{volume}} = 1 - I_{\text{density}}$$

Similarly, the influence of molecular composition on sinking velocity was decoupled into those of water and dry content (proteins, lipids, carbohydrates, and others). In this case, the sinking velocity was rewritten as a function of water volume, dry volume, and dry density (taking water density as a constant).

$$v_{sink} = f(V_{water}, V_{dry}, \rho_{dry})$$

The first-order Taylor expansion near the high-nutrient condition $\left(V^0_{water}, V^0_{dry}, \rho^0_{dry}\right)$ indicates the influences of water volume, dry volume, and dry density. The influence of each variable, $I_{water}$, $I_{dry\ volume}$, or $I_{dry\ density}$, is the ratio of each derivative term to the total of all three terms. For example, $I_{water}$ can be rewritten as:

$$I_{water} = \frac{\frac{\partial f\left(V^0_{water}, V^0_{dry}, \rho^0_{dry}\right)}{\partial V_{water}} \cdot \Delta V_{water}}{\frac{\partial f\left(V^0_{water}, V^0_{dry}, \rho^0_{dry}\right)}{\partial V_{water}} \cdot \Delta V_{water} + \frac{\partial f\left(V^0_{water}, V^0_{dry}, \rho^0_{dry}\right)}{\partial V_{dry}} \cdot \Delta V_{dry} + \frac{\partial f\left(V^0_{water}, V^0_{dry}, \rho^0_{dry}\right)}{\partial \rho_{dry}} \cdot \Delta \rho_{dry}}$$

The influence of dry content can be further calculated as the sum of dry volume and dry density as:

$$I_{dry} = I_{dry\ volume} + I_{dry\ density}$$

## Phytoplankton species, culture conditions, and drug treatments

All phytoplankton species were obtained from Provasoli–Guillard National Center for Marine Algae and Microbiota and the species belong to the Culture Collection of Marine Phytoplankton (CCMP). All species identifications were verified using light microscopy, as well as TEM for a subset of species. A list of species, their CCMP identifiers, and their maintenance growth media are shown in S3 Table.

All algae were cultured as previously [32]. High nutrient condition corresponds to L1 (or L1-Si) media, low nutrient condition corresponds to the high nutrient conditions with the nutrients being diluted 100-fold, and specific nutrient starvations correspond to the high nutrient condition in the absence of the indicated nutrient. For experiments, maintenance cultures were split 1:4 for 2 days to achieve exponentially growing cultures. These cultures were then split to each indicated nutrient condition and culture for 5 days prior to measurements of cell sinking. The lighting was available from a nearby window at approximately 50 µmol/s/m$^2$ of PAR and day length varying between 9 and 14 h. All cultures were grown at 22 °C ± 1.5 °C. Key experiments were repeated using a controlled lighting setup with 100 µmol/s/m$^2$ of PAR with day length set at 12 h and temperature at 22 °C.

Lipid synthesis was inhibited with cerulenin (Cayman Chemical, Cat#10005647) and C75 (Cayman Chemical, Cat#10005270). Lipid synthesis inhibitors were used at 10 µM concentration and the treatment duration corresponded to the low nutrient starvation (5 days).

## Cell buoyant mass and volume measurements

Cell buoyant mass and volume measurements were carried out identically to previous report [32]. In short, buoyant masses were measured using the SMR [54], where a vibrating cantilever with a microfluidic channel detects changes in resonant frequency due to the presence of a cell, which correlates to its buoyant mass. Two SMRs were used: one with an 8 × 8 µm cross-section for smaller phytoplankton cells and one with a 15 × 20 µm cross-section for larger cells. Cell volumes were measured using a Coulter counter (Multisizer 4), based on electric impedance changes as cells passed through an orifice. Two aperture sizes (20 and 100 µm) were used depending on cell size. Both SMR and Coulter counter provide measurements of single cells within 1–100 ms per cell, and the measurements were carried out at room

temperature in L1-Si/100 media. The measurements were calibrated using NIST-certified polystyrene beads (2–10 µm, Duke Standards, Thermo Scientific). All measurements were carried out between 9:30AM and 3:30 pm, and paired samples (*e.g.*, high nutrient and corresponding low nutrient sample) were measured within 1 hour of each other.

### Determining cell sinking velocities and Péclet numbers

Phytoplankton gravitational sinking velocities, $v_{sink}$, were calculated according to Eq (1). The following values were used for environmental constants: the dynamic viscosity of seawater of $1.07 \times 10^{-3}$ Pa·s, and density of seawater of 1.026 g/ml. Population average cell radius was calculated from the cell volume measurements, and population average cell density was calculated by comparing the volume and buoyant mass measurements. All phytoplankton species in this study were estimated to have low Reynolds numbers ($<10^{-6}$). We note that the exact gravitational sinking velocities would be influenced by changes in seawater viscosity and density, which could occur if, for example, seawater temperature changed significantly.

Péclet numbers were calculated according to $Pe = UL/D$, where $U$ is the sinking velocity, $L$ is the cell length, and $D$ is the external particle's diffusivity [55]. We used the diffusivity value of 1,700 µm2/s for nitrate and 10 µm$^2$/s for a small particle (*e.g.*, viral particle).

### Cell proliferation rate measurements

Cell proliferation rates were derived from Coulter counter-based cell count measurements. Cell counts were measured on three consecutive days, where the middle day corresponded to the cell mass and volume measurements. Proliferation rates were calculated assuming exponential growth.

### Fluorescent labeling of proteins, lipids, and DNA

For all fluorescent labeling approaches, the cells were fixed for 10 min in L1-Si media (or corresponding nutrient-limited media) containing 4% formaldehyde, after which the cells were washed twice with phosphate-buffered saline (PBS). Neutral lipids were stained using 2 µM Bodiby 493/503 (4,4-Difluoro-1,3,5,7,8-Pentamethyl-4-Bora-3a,4a-Diaza-s-Indacene, ThermoFisher, Cat#D3922) in PBS for 20 min. After staining, the cells were washed two times with PBS. Cellular proteins were stained using the amine-reactive LIVE/DEAD Fixable Red Dead Cell Stain (ThermoFisher, Cat#L34972) using 2× supplier-recommended concentration in PBS for 15 min [56]. After staining, the cells were washed with PBS solution containing 5% Bovine Serum Albumin, and again with PBS. DNA was stained using 10 µg/ml Hoechst 33342 (ThermoFisher, Cat#H3570) in PBS for 20 min. After staining, the cells were washed two times with PBS.

### Fluorescence and brightfield microscopy

Fluorescence and brightfield microscopy samples were plated on poly-L-lysine (Sigma-Aldrich, Cat#P8920) coated glass-bottom CELLVIEW plates (Greiner Bio-One) for >30 min prior to imaging. The samples were imaged at RT using DeltaVision wide-field deconvolution microscope with standard DAPI, FITC, TRICT, Cy5, and POL filters, and a 100× oil-immersion objective. z-layers were typically collected with 0.3 µm spacing covering >8 µm in height. Fluorescence image deconvolution was carried out using DeltaVision software.

### Flow cytometry

The flow cytometry samples were identical in preparation to those used for fluorescence microscopy. Instead of plating on imaging plates, the samples were cleaned from aggregates using a 100 µm filter. The samples were then analyzed using BD Biosciences FACS Celesta with 405, 488, and 561 nm excitation lasers and 450/40 nm, 515/20 nm, 610/20 nm, and 710/50 nm emission filters. Samples were gated on FSC and SSC to exclude too small particles, and a typical analysis

 

measured 10,000 cells within the FSC/SSC gate. Chlorophyl autofluorescence was used to ensure that all measured particles were phytoplankton cells.

## Water and dry content measurements

Cellular water and dry content, including the density and volume of cell's dry contents, were determined as detailed before [32]. In short, the average buoyant mass of cells in a given population was measured with the SMR, as detailed above. These measurements were then repeated after moving the cells to L1-Si/100 media where 90% of the water was deuterated ($D_2O$). $D_2O$ mixes freely with the water inside the cell and $D_2O$ has a higher density than $H_2O$, resulting in a different population average buoyant mass. By comparing these two buoyant mass averages, along with the measurement solution densities (L1-Si$_{H2O}$ and L1-Si$_{D2O}$), we can solve for the average dry volume and the density of the dry volume in the cell population [57,58]. These measurements were then compared to the Coulter counter-based total cell volume measurements to derive the volume of water inside the cells.

## Electron microscopy

For TEM sample preparation, the cells were fixed for 60 min in L1-Si media (or corresponding nutrient-limited media) containing glutaraldehyde and paraformaldehyde at final concentrations of 2.5% and 2%, respectively. After fixation, the cells were washed two times with 100 mM sodium cacodylate (pH 7.2). After washing, samples were fixed for 60 min on ice with 1% osmium tetroxide in a solution containing 1.25% potassium ferrocyanide and 100 mM sodium cacodylate. Next, the samples were washed multiple times with 100 mM sodium cacodylate and with 50 mM sodium maleate. The samples were then stained with 2% uranyl acetate in sodium maleate buffer at RT o/n. The samples were rinsed with DI water and dehydrated using a series of 10 min ethanol incubations. The ethanol concentrations varied in ascending order, from 30% to 100%. The samples were then washed twice in propylene oxide for 30 min, and moved to propylene oxide—resin mixture (1:1) for o/n. The samples were moved to a new propylene oxide—resin mixture (1:2) for 6 hours and then into pure resin o/n. The samples were then moved to molds and polymerized at 60 °C for 48 hours to form blocks suitable for sectioning. Thin sections of 60 nm were obtained using a Leica UC7 ultramicrotome. The sections were collected on carbon-coated nitrocellulose film copper grids.

TEM imaging was carried out using FEI Tecnai T12 transmission electron microscope and an AMT XR16 CCD camera. Typical imaging was carried out with 120 kV voltage. Imaging magnification varied between experiments. All TEM images shown in the manuscript are representative examples.

For SEM sample preparation, the cells were plated on poly-L-lysine-coated cover slides for 40 min prior to fixation. The fixatives, 2.5% glutaraldehyde and 2% paraformaldehyde, were added to the cells on the cover slides for 60 min. The samples were then washed two times with 100 mM sodium cacodylate (pH 7.2) and post-fixed for 30 min at +4 °C using 1% osmium tetroxide in sodium cacodylate buffer. The samples were rinsed with DI water and dehydrated using a series of 10 min ethanol incubations. The ethanol concentrations varied in ascending order from 35% to 100%. Next, the samples were incubated with ethanol/tetramethyl silane mixtures (50/50 mixture, and 20/80 mixture, respectively) for 15 min each. Then, the samples were washed twice with 100% tetramethyl silane. The wash solution was removed, and the samples were left to dry o/n. The dried samples were sputter-coated with gold.

SEM imaging was carried out using Zeiss Crossbeam 540 scanning electron microscope. Typical imaging was carried out with 4 kV accelerating voltage, 600 pA probe current, a working distance of 8 mm, and a magnification of 4,000×. All SEM images shown in the manuscript are representative examples.

## Image analysis

Lipid droplets, starch granules, and acidocalcisomes were visually identified from TEM images based on existing literature [37,59,60]. For quantifications, cells, along with intracellular components of interest, were segmented and quantified using

MATLAB (R2023a). For each cell, the cell boundary, lipid droplets, starch granules, and acidocalcisomes were manually defined by a freehand line profile drawn using MATLAB's Image Processing and Computer Vision toolbox. Intracellular composition was then estimated using the area of a component of interest relative to the total cell area. The script can be found at https://github.com/alicerlam/algae and at https://doi.org/10.5281/zenodo.17478443.

## Statistics

In all figures, *N* refers to the number of independent cultures, *n* refers to the number of separate cells measured. Particles too small to be viable cells were removed from all final analyses. Statistical tests are detailed in figure legends. All *p*-values were calculated using OriginPro (2025) software.

## Supporting information

**S1 Fig. Simulations of cell sinking are largely independent of starting cell composition. (A–C)** Simulations of cell sinking velocity in a typical green alga (A, *Dunaliella tertiolecta*), diatom (B, *Chaetoceros calcitrans*), and a hypothetical average species (C) as a function of changing cell composition. Sinking velocity changes due to cell volume and cell density changes are separated in blue and orange, respectively, and the total cell sinking change is depicted by a red dotted line. Changes in dry content refer to corresponding changes in all other contents except water. For large water content increases, separation of volume and density contributions are excluded. See S1 and S2 Tables for more details. Simulation data can be found in the S1 Data file.
(TIF)

**S2 Fig. Single-cell mass and volume responses to starvation. (A)** Schematic of workflow for measuring single-cell buoyant masses and volumes, which were then used to determine population average density, volume, and gravitational sinking velocity. SMR is a microfluidic device, where a cell is flown through a channel embedded in a vibrating cantilever. The change in the cantilever's vibration frequency is proportional to the buoyant mass of the cell. Coulter counter is a fluidic device, where a cell is flown through a small aperture, displacing the measurement solution (artificial seawater). This changes the electrical resistance across the aperture in a manner proportional to cell volume. **(B)** Representative single-cell buoyant mass (top) and volume (bottom) histograms following a 5-day culture under high (gray) and low (orange) nutrient conditions in indicated species. All data can be found in the S1 Data file.
(TIF)

**S3 Fig. Phytoplankton cell volumes and densities change when starved, but cell shapes do not. (A)** Cell volume (top row), density (middle row), and proliferation rate (bottom row) under indicated nutrient conditions in indicated species. Proliferation rates reflect the number of cell doublings/day, and these were calculated from cell counts measured on days 4, 5, and 6, following starvation start (cell mass and volume measurements were carried out on day 5). Dots depict independent cultures, bars and whiskers depict mean ±SD, *N* depicts the number of independent cultures. Note that the *Emiliania huxleyi* strain is non-calcifying. **(B)** Representative fluorescence microscopy images of indicated species following a 5-day culture under indicated nutrient conditions. The cells were labeled for total protein content (red-to-yellow) and DNA content (blue-to-white). Cell outlines are highlighted with a dotted line for one cell in each fluorescence image. Brightfield images were used instead of fluorescence imaging for *Heterosigma akashiwo* and *Isochrysis galbana*. All scalebars denote 5 μm. Note that *H. akashiwo* is zoomed out 2× in comparison to other images. *n* > 20 cells per condition. *Phaeodactylum tricornutum* cells were near-exclusively in the fusiform morphotype. All data can be found in the S1 Data file.
(TIF)

**S4 Fig. Starvation-induced cell sinking velocities are reversible.** Gravitational sinking velocities were measured following 5-day starvation under low nutrient condition after which nutrients were resupplied and sinking velocities were measured again 2 days later (rescue). All data are normalized to high nutrient condition on day 5. *p*-values depict Welch's *t* test between low nutrient and rescue conditions. All data can be found in the S1 Data file.
(TIF)

**S5 Fig. Cellular dry and water content following starvation. (A)** Volume of water (top row) and dry contents (middle row), and the density of the dry contents (bottom row) in an average cell in indicated species following a 5-day culture under indicated starvation conditions. *N* depicts the number of independent cultures (dots), bar and whiskers depict mean±SD. **(B)** The relative influence of cellular water volume (blue), dry volume (green), and the density of dry content (orange) changes on cell sinking velocity changes under low nutrient starvation condition. Data depicts mean±SEM. All data can be found in the S1 Data file.
(TIF)

**S6 Fig. Lipid and protein content changes following starvation. (A)** Relative changes in cellular neutral lipid content following 5-day culture under indicated starvation conditions. Data normalized to high nutrient condition within each species. *N* depicts the number of independent cultures (dots), bar and whiskers depict mean±SD. **(B)** Same as (A), but data is for cellular protein content. Data for *Phaeodactylum tricornutum* is a repeat of the data shown in Fig 5A and 5D. See S3B Fig for example fluorescence images of the cellular protein content labeling. All data can be found in the S1 Data file. Note that the data represent neutral lipid and protein content per cell, but as many species increase their cell volume following starvation, the concentration of neutral lipids and proteins may differ significantly from the changes visualized here.
(TIF)

**S7 Fig. *Chlamydomonas* sp. accumulates lipid droplets exclusively at the cell periphery. (A)** Fluorescence microscopy of neutral lipids in *Chlamydomonas* sp. cells following 5-day culture under high and low nutrient conditions. Both maximum intensity and single z-layer images are shown to highlight the lipid droplet localization at cell periphery. Scale bars denote 10 μm. **(B)** TEM of *Chlamydomonas* sp. cells following 5-day culture under high and low nutrient conditions. Zoom-ins (bottom row) visualize lipid droplets. Key cell compartments are indicated with letters (C, chloroplast; L, lipid droplet; N, nucleus; S, starch granule). Scale bars denote 1 μm on the top row and 200 nm on the bottom row (zoom-ins). **(C, D)** Quantifications of lipid droplet diameter (C) and count (D) from TEM images, *p*-value depicts Welch's *t* test. **(E)** Scanning electron microscopy (SEM) of *Chlamydomonas sp*. cells. Cell dehydration and shrinkage during SEM sample prep reveals "bumps" on the cell surface, which are identical in size and location to the lipid droplets observed in TEM and fluorescence microscopy. Scalebars denote 1 μm. All data can be found in the S1 Data file.
(TIF)

**S8 Fig. Starving green algae accumulate larger starch granules and, in some cases, acidocalcisome-like organelles. (A)** TEM imaging of *Chlamydomonas* sp. following 5-day culture under nitrogen and phosphorus starvation. Scalebars depict 1 μm. Key cell compartments are indicated with letters in the zoom in (A, acidocalcisome; C, chloroplast; L, lipid droplet; M, mitochondria; S, starch granule). **(B, C)** Quantifications of average starch granule area (B) and number per cell (C) from TEM images. Dots depict separate cells, *n* depicts number of cells analyzed, bar and whiskers depict mean±SD. **(D)** Quantifications of acidocalcisome number per cell from TEM images. Acidocalcisomes are electron-dense, round organelles in TEM images, which often store calcium, phosphate, and metals. Analysis was limited to *Chlamydomonas* sp., as acidocalcisomes were not observed in other green algae. Dots depict separate cells, *n* depicts number of cells analyzed, bar and whiskers depict mean±SD. **(E)** Simulated influence of water, starch, protein, and lipid content changes

on cell sinking velocity in the green algae species. The analysis was limited to comparing high and low nutrient conditions, where cellular content changes were determined most comprehensively. Data is normalized so that the components add up to 100%. Note that for *Chlamydomonas* sp. the starch accumulation can counteract sinking velocity influence of lipid accumulation and protein loss, but additional mechanism(s) must be present to fully explain the experimentally determined cell sinking. In panels B–D, *p*-values were obtained using ANOVA followed by Tukey's posthoc test. All data can be found in the S1 Data file.
(TIF)

**S1 Table. Molecular (dry) density values used in simulations.**
(PDF)

**S2 Table. Cell composition values used in simulations.** The fractional (w/w) dry content values are derived from literature and the fractional water content values are derived from experiments in this paper.
(PDF)

**S3 Table. Details of the phytoplankton species studied.**
(PDF)

**S1 Data. Raw numerical data underlying each figure.** See readme on the first datasheet for more details.
(XLSX)

## Acknowledgments

We thank the Koch Institute's Robert A. Swanson (1969) Biotechnology Center, specifically the Peterson (1957) Nanotechnology Materials Core Facility (RRID: SCR_018674), the Microscopy Core Facility, and the Flow Cytometry Core Facility for their support. We also thank Prof A. Babbin for useful feedback.

## Author contributions

**Conceptualization:** Yanqi Wu, Teemu P. Miettinen.

**Data curation:** Yanqi Wu, Vieyiti K. Kouadio, Teemu P. Miettinen.

**Formal analysis:** Yanqi Wu, Thomas R. Usherwood, Reshum Aurora, Aaron Z. Lam, Alice R. Lam, Teemu P. Miettinen.

**Funding acquisition:** Abigail K. R. Lytton-Jean, Scott R. Manalis.

**Investigation:** Yanqi Wu, Vieyiti K. Kouadio, Thomas R. Usherwood, Justin Li, Margaret Bisher, Abigail K. R. Lytton-Jean, Teemu P. Miettinen.

**Methodology:** Yanqi Wu, Thomas R. Usherwood, Margaret Bisher, Teemu P. Miettinen.

**Project administration:** Teemu P. Miettinen.

**Resources:** Abigail K. R. Lytton-Jean, Teemu P. Miettinen.

**Supervision:** Scott R. Manalis, Teemu P. Miettinen.

**Validation:** Yanqi Wu, Teemu P. Miettinen.

**Visualization:** Yanqi Wu, Teemu P. Miettinen.

**Writing – original draft:** Yanqi Wu, Teemu P. Miettinen.

**Writing – review & editing:** Yanqi Wu, Vieyiti K. Kouadio, Thomas R. Usherwood, Justin Li, Scott R. Manalis, Teemu P. Miettinen.

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
