## [Editor Report · Decision Letter 0]

9 Sep 2025

Dear Dr Miettinen,

Thank you for submitting your manuscript entitled "Biophysical and molecular mechanisms responsible for phytoplankton sinking in response to starvation" for consideration as a Research Article by PLOS Biology.

Your manuscript has now been evaluated by the PLOS Biology editorial staff, as well as by an academic editor with relevant expertise, and I'm writing to let you know that we would like to send your submission out for external peer review.

Once your full submission is complete, your paper will undergo a series of checks in preparation for peer review. After your manuscript has passed the checks it will be sent out for review. To provide the metadata for your submission, please Login to Editorial Manager (https://www.editorialmanager.com/pbiology) within two working days, i.e. by Sep 11 2025 11:59PM.

Kind regards,

Roli Roberts

Roland Roberts, PhD

Senior Editor

PLOS Biology

rroberts@plos.org

---

## [Decision Letter · Decision Letter 1]

21 Oct 2025

Dear Dr Miettinen,

Thank you for your patience while your manuscript "Biophysical and molecular mechanisms responsible for phytoplankton sinking in response to starvation" was peer-reviewed at PLOS Biology. It has now been evaluated by the PLOS Biology editors, an Academic Editor with relevant expertise, and by two independent reviewers.

Based on the reviews, we are likely to accept this manuscript for publication, provided you satisfactorily address the remaining points raised by the reviewers, and the following data and other policy-related requests.

IMPORTANT - please attend to the following:

a) Please change your Title to something with an active verb. We suggest: "Diverse biophysical and molecular mechanisms drive phytoplankton sinking in response to starvation"

b) Please address the requests from the reviewers (see comments at the foot of this email).

c) Please address my Data Policy requests below; specifically, we need you to supply the numerical values underlying Figs 1C, 2BCD, 3ABC, 4, 5ADEFG, 6BC, S1ABC, S2B, S3A, S4, S5AB, S6AB, S7CD, S8BCDE, either as a supplementary data file or as a permanent DOI’d deposition. I also note that you already have a supplementary file (Tables S4-S9); please could re-name this “S1_Data.xlsx” and clarify its relationship to the Figures?

d) Please cite the location of the data clearly in all relevant main and supplementary Figure legends, e.g. “The data underlying this Figure can be found in S1 Data” or “The data underlying this Figure can be found in https://zenodo.org/records/XXXXXXXX

e) Please make any custom code available, either as a supplementary file or as part of your data deposition.

We expect to receive your revised manuscript within two weeks.

*Published Peer Review History*

*Press*

Sincerely,

Roli Roberts

Roland Roberts, PhD

Senior Editor

rroberts@plos.org

PLOS Biology

DATA POLICY:

Regardless of the method selected, please ensure that you provide the individual numerical values that underlie the summary data displayed in the following figure panels as they are essential for readers to assess your analysis and to reproduce it: Figs 1C, 2BCD, 3ABC, 4, 5ADEFG, 6BC, S1ABC, S2B, S3A, S4, S5AB, S6AB, S7CD, S8BCDE. NOTE: the numerical data provided should include all replicates AND the way in which the plotted mean and errors were derived (it should not present only the mean/average values).

CODE POLICY

DATA NOT SHOWN?

REVIEWERS' COMMENTS:

Reviewer #1:

[identifies herself as Amy Ikui]

Wu et al. conducted an elegant and careful study that combines biophysical modeling with cell measurements to investigate how nutrient starvation affects their sinking behavior of marine phytoplankton. They found that most species sink faster under starvation, although the underlying mechanisms vary among taxa. In some species, accelerated sinking results from higher cell density, whereas in others it reflects changes in cell volume. Notably, lipid accumulation as observed in Phaeodactylum tricornutum, decreases cell density and slows sinking velocity. Starch accumulation is also involved in the sinking velocity under starvation in green algae species. These findings demonstrate that phytoplankton employ diverse biophysical and molecular strategies to regulate buoyancy, thereby facilitating vertical migration and contributing to oceanic carbon flux under nutrient-limited conditions. The strength of this manuscript is its comparative analysis across nine marine phytoplankton species, which allows the authors to dissect mechanistic insights to biological significance in an area that remains under studied.

Overall, the manuscript is clearly written and logically organized. The research motivation is compelling, and the authors present comprehensive data to support their conclusion. I strongly recommend this manuscript for publication in PLOS Biology.

Minor points:

Line 79: (Chartoceros calcitrans in FigS1B or Phaeodactylum tricornutum in Fig1C)

Line 112: Remove extra "that"

Line 113: Isochrysis galbana showed no change in velocity. If measurements of dry volume, density, and lipid/protein/water content, are already available, it would be interesting to compare these with the other two haptohytes. Such comparisons could reveal whether compensatory changes among cellular components account for the lack of change in estimated sinking velocity.

Figures S5A and S6: Consider moving some of these data to Figure 4, if there are sufficient space to include them alongside the modeled values.

Fig 2D: If Y axis represents each species, why the bar shows slope. At first, I thought the slope has analytical meaning, then I realize that this is probably a presentation adjustment to avoid overlap among the bars. In any cases, an explanation would help. Alternatively, consider plotting the bars in parallel to simplify interpretation.

Figure 3A is a summary of all the data, but it is somewhat difficult to see. You may change the background color or label contrast, which will improve readability.

Fig S3A: Provide a little more clarity on calculation of proliferation rates and the exact meaning of the numbers in Y axis, in the methods section and/or in figure legends and graph axis titles.

Fig S3B Legend: "Brightfield images were used for instead of fluorescence imaging" should be "Brightfield images were used instead of fluorescence imaging"

Fig S4 appears before Fig 3B in the text in page 6.

Reviewer #2:

This paper provides a thorough quantitative investigation of sinking velocities upon starvation across 9 phytoplankton species. High quality measurements of cell volume and density using a state-of-the art technique (SMR) are combined with an interesting model which allows deciphering the respective contribution of volume increase vs. density changes in cell sinking. In the last part of the paper, the molecular changes driving cell density variations (i.e., lipid vs. starch accumulation) are investigated by combining EM imaging and drug treatments. Overall, this is a high-quality quantitative investigation; the model provides interesting insights, the experimental measurements are of high-quality and the findings raise interesting future research directions. I fully recommend this paper for publication.

I only have few recommendations, mostly aimed at easing the reader's understanding.

1) The paper sometimes skips important methodology description which would make the findings more convincing and easier to understand. (i) For example, results in Figure 3 -B are really interesting (and nicely displayed) but it is not possible to understand how they were obtained from reading the main text (how the respective influence of cell volume vs. density on sinking velocity was calculated). The model is a strength of this paper and it could be better explained in the main text. (ii) Another example is that references 32-34 are important to understand how data were acquired/quantified and 1-2 sentences summarizing what these previous findings were would be helpful.

2) The discussion is interesting but perhaps a bit dense. (i) The first paragraph emphasizes the interest of the model which provides estimates for how buoyancy could be achieved and brings an interesting perspective with respect to recent publication on Pyrocystis noctiluca. (ii) The second discussion on cell size increase is a bit confusing since one claim from the paper is that density - not volume - is the main parameter driving sinking velocity changes. It thus seems counter-intuitive to relate the *small* changes in volume to arguments of cell surface area/resource intake. (iii) The fourth paragraph rightfully clarifies that sinking may not be directly regulated but a byproduct of response to strvation. This statement is needed since some of the phrasing earlier in the paper suggests the opposite (i.e., 'phytoplankton increase sinking velocity in response to starvation' ). However, the paragraph mixes a lot of ideas and is not easy to understand for non-specialists.

---

## [Editor Report · Decision Letter 2]

6 Nov 2025

Dear Dr Miettinen,

Thank you for the submission of your revised Research Article "Diverse biophysical and molecular mechanisms drive phytoplankton sinking in response to starvation" for publication in PLOS Biology. On behalf of my colleagues and the Academic Editor, Holly Bik, I'm pleased to say that we can in principle accept your manuscript for publication, provided you address any remaining formatting and reporting issues. These will be detailed in an email you should receive within 2-3 business days from our colleagues in the journal operations team; no action is required from you until then. Please note that we will not be able to formally accept your manuscript and schedule it for publication until you have completed any requested changes.

Sincerely, 

Roli Roberts

Senior Editor

PLOS Biology

rroberts@plos.org